# Mechanical Behaviors of Si/CNT Core/Shell Nanocomposites under Tension: A Molecular Dynamics Analysis

**DOI:** 10.3390/nano11081989

**Published:** 2021-08-02

**Authors:** Jee Soo Shim, Gi Hun Lee, Cheng Yu Cui, Hyeon Gyu Beom

**Affiliations:** Department of Mechanical Engineering, Inha University, 100 Inha-ro, Incheon 22212, Korea; dogfin8@inha.edu (J.S.S.); solafide27@gmail.com (G.H.L.); cuisw92@hanmail.net (C.Y.C.)

**Keywords:** core/shell nanocomposite, fracture mechanism, mechanical property, silicon nanowire, carbon nanotube, molecular dynamics

## Abstract

The silicon/carbon nanotube (core/shell) nanocomposite electrode model is one of the most promising solutions to the problem of electrode pulverization in lithium-ion batteries. The purpose of this study is to analyze the mechanical behaviors of silicon/carbon nanotube nanocomposites via molecular dynamics computations. Fracture behaviors of the silicon/carbon nanotube nanocomposites subjected to tension were compared with those of pure silicon nanowires. Effective Young’s modulus values of the silicon/carbon nanotube nanocomposites were obtained from the stress and strain responses and compared with the asymptotic solution of continuum mechanics. The size effect on the failure behaviors of the silicon/carbon nanotube nanocomposites with a fixed longitudinal aspect ratio was further explored, where the carbon nanotube shell was found to influence the brittle-to-ductile transition behavior of silicon nanowires. We show that the mechanical reliability of brittle silicon nanowires can be significantly improved by encapsulating them with carbon nanotubes because the carbon nanotube shell demonstrates high load-bearing capacity under tension.

## 1. Introduction

The development of high-capacity lithium-ion batteries (LIBs) with enhanced cycling stability and reliability has attracted considerable attention due to the potential applications of these batteries in the field of electric vehicle manufacturing [1]. Silicon (Si) electrode with a high specific capacity of 4200 mAhg^−1^ is theoretically considered to be the most promising component of LIBs [2]; nevertheless, the degradation of the mechanical reliability of LIBs, resulting from pulverization during lithiation/delithiation processes, remains a significant challenge. Upon repeated charging of LIBs, lithium ions penetrate the Si electrode, which induces catastrophic failure of the electrode material [3,4]. To solve these problems, various researchers have attempted to fabricate Si nanostructures with improved mechanical strength and fracture toughness; however, the miniaturization technique could not fully prevent the pulverization process [5].

Another way to enhance the mechanical reliability of LIBs is to design a core/shell nanocomposite using carbon-based materials, for example, graphene, fullerene, and carbon nanotubes (CNTs). Excellent mechanical properties and electronic conductivity of carbon-based materials were found to improve the performance of electrode materials [6,7,8]. Via chemical vapor deposition, Son et al. [9] fabricated a nanocomposite anode consisting of a graphene–silica assembly and realized a charge capacity retention of 78.6% after 500 cycles. Bae [10] and Shao et al. [11] experimentally designed a core/shell nanocomposite anode comprising Si and CNTs. They showed that the pulverization phenomenon observed in the Si anode was effectively delayed using the as-fabricated core/shell nanocomposite anode. In addition, it was reported that the Si/CNT core/shell nanocomposites were successfully fabricated by depositing graphene on Si nanowires (SiNWs) [12,13]. Therefore, the Si/CNT core/shell nanocomposite electrode model is promising for application in LIBs.

Nevertheless, the key factors that enhance the mechanical properties of the electrode materials in a core/shell nanocomposite system have not been elucidated to date. Deformation behavior of core/shell nanocomposite electrodes should be analyzed to better understand the higher reliability of core materials. This will lead to an improvement in the fabrication techniques of nanostructures. However, conventional experimental testing or continuum-based modeling schemes are limited to the studies of the atomic-level physics of deformations, such as slip and twinning plastic deformation [14]. Furthermore, note that the deformation behaviors of nanomaterials are different from those of their bulk counterparts with different surface-to-volume ratios [15]; consequently, the surface-to-volume ratio influences the size dependence of mechanical properties at the nanoscale [16].

In this context, Zang and Zhao [17] conducted molecular dynamics (MD) simulations to investigate the buckling properties of Si/CNT (core/shell) nanocomposites and found that the strengthening mechanism of the nanocomposites resulted from the phase transformation of Si. Wu et al. [18] examined the torsional and bending properties of Si/CNT nanocomposites via MD simulations and discovered that the interwall interactions enhanced these properties. Nevertheless, to the best of our knowledge, to date, mechanical properties and deformation mechanisms of the Si/CNT nanocomposites subjected to tension have not been completely analyzed from an atomistic fracturing viewpoint. Under tension, SiNWs exhibit a complex failure mechanism because of their crystallographic anisotropy [16,19]. Fracture properties of the encapsulated SiNWs can be further influenced by the high load-bearing capacity of CNT shells [20].

Herein, the key factors that can improve the mechanical reliability of SiNWs in core/shell nanocomposites were investigated. MD simulations of the axial tensile test were performed on pure SiNWs and Si/CNT (core/shell) nanocomposite models. To explore the strengthening mechanisms for the brittle and ductile failures of SiNWs, two types of Si/CNT nanocomposite models were examined: (1) [111]-oriented SiNWs encapsulated with CNTs and (2) [110]-oriented SiNWs encapsulated with CNTs, which were designated as “Model-1” and “Model-2”, respectively. Detailed information about the strengthening mechanism of mechanical properties is provided based on an atomistic view of fracture. Asymptotic solutions for the core/shell nanocomposite models were formulated according to continuum mechanics. The size effect on the Young’s modulus (*Y*) of Si/CNT nanocomposites with a fixed longitudinal aspect ratio is discussed.

## 2. Computational Methodology of MD

### 2.1. Interatomic Potential

MD simulations predict the kinetic dynamics of atoms by numerically solving Newton’s equation of motion [21], where the choice of correct interatomic potentials is crucial. In this study, three types of interatomic potential functions were employed to model the core/shell nanocomposites. The potential energy for Si single crystals was reproduced using the modified embedded atom method (MEAM) potential [22], which has been reported to appropriately characterize the covalent bonds of the diamond cubic phase of Si [23]. A cutoff distance of 0.6 nm was used to simulate the correct fracture properties of Si [19,24].

The potential energy for CNTs was calculated by the 2nd generation reactive empirical bond order (REBO) potential, which can be expressed as [25,26]
(1)$$E^{\mathrm{REBO}}=\sum_{i}\sum_{j<i}\left[E_{R}(r_{ij})+\kappa_{ij}E_{A}(r_{ij})\right],$$
where
(2)$$E_{R}\left(r_{ij}\right)=f_{ij}\left(1+Q_{ij}/r_{ij}\right)H_{ij}e^{-\alpha_{ij}r_{ij}}$$
(3)$$E_{A}(r_{ij})=-f_{ij}\sum_{n=1,3}b_{ij}^{\left(n\right)}e^{-\beta_{ij}^{\left(n\right)}r_{ij}},$$
where $r_{ij}$ indicates the interatomic distance between the atoms *i* and *j*; $E_{R}$ and $E_{A}$ are functions that represent repulsive and attractive interactions, respectively, which are coupled by the bonding function $\kappa_{ij}$; $Q_{ij}$, $H_{ij}$, $\alpha_{ij}$, $b_{ij}$, and $\beta_{ij}$ are model constants [25]; $f_{ij}$ is the cutoff function, where the switching term was removed to exclude the abnormal strain hardening of CNTs [27]; and the cutoff distance was set at 0.2 nm. The material properties of the SiNWs and CNTs were investigated via the MD simulations (based on the selected potentials) and were consistent with the experimental results, as presented in Table 1.

The Lennard–Jones (LJ) potential was employed to reproduce the atomic interactions between the SiNWs and the CNTs, which is expressed as [32]
(4)$$E_{ij}^{LJ}=4\zeta \left[{\left(\psi /r_{ij}\right)}^{12}-{\left(\psi /r_{ij}\right)}^{6}\right]$$
where ζ and ψ were fixed at 0.00347 eV and 0.3764 nm, respectively [33], and the cutoff distance for the Si–C atomic bonds was measured to be 1.2 nm.

### 2.2. Simulation Procedure

Tensile testing of the Si/CNT nanocomposite models with a fixed longitudinal aspect ratio was conducted in the MD framework. The atomistic model of a representative Si/CNT nanocomposite is shown in Figure 1. Circular SiNWs with a diamond cubic lattice structure were wrapped by armchair CNTs. Both structures showed concentric longitudinal axes. The axial orientation of the simulation models corresponded to the *z*-direction in the Cartesian coordinate system. The SiNWs and CNTs demonstrated identical lengths along the *z*-direction, and the lattice translational indices (LTIs) of the CNTs used herein are presented in Table 2. The thickness of the CNT shell was 0.34 nm, which was determined based on the effective range of van der Waals interactions [34].

The longitudinal aspect ratio (length/diameter) of the Si/CNT nanocomposites was fixed at approximately 3.1. Accordingly, the CNT shell with the (25,25) LTI encapsulated the SiNWs with a length of 9 nm and a diameter of 2.9 nm. Various dimensions were selected for the core/shell nanocomposites according to previous studies [35,36]. All simulation models were subjected to a non-periodic boundary condition in three Cartesian directions. The deformation mechanism of SiNWs depends on the crystallographic system [19]. Therefore, in this study, the [111]- and [110]-oriented SiNWs were examined. We designated the core/shell nanocomposites consisting of [111]-, and [110]-oriented SiNWs with CNTs as ‘Model-1′ and ‘Model-2′, respectively.

Based on the interatomic potentials, relaxed molecular structures were obtained via the conjugate gradient method [37]. To analyze the fracture behavior of the models at the atomic level, the temperature was set to 1 K using a Nosé–Hoover thermostat [38]. The unit time step was fixed at 1 fs, and a strain rate of 5 × 10^8^ s^−1^ was applied in the *z*-direction. Herein, the average stress in the simulation models was calculated according to the virial stress that corresponds to the Cauchy stress of continuum mechanics [39]. All MD simulation procedures were performed using the Large-scale Atomic/Molecular Massively Parallel Simulator code [21], and atomistic images were obtained using the OVITO software [40].

## 3. Continuum Interpretation of the Core/Shell Nanocomposite under Tension

A core/shell composite structure was subjected to tensile loading along the *z*-direction. The origin of the cylindrical coordinate system (r,θ,z) was located at *O* (Figure 1). The radii of the core and shell materials were denoted as $r_{1}$ and $r_{2}$, respectively. For a linear elastic model of the Si/CNT nanocomposite, the core and shell materials are assumed to satisfy the cylindrically orthotropic and isotropic conditions, respectively. Mechanical response of the thin shell structure was investigated based on the membrane theory of continuum mechanics [41].

Stress and strain relations for the core and shell materials can be expressed as
(5)$$\sigma_{r}^{\left(1\right)}=\sigma_{\theta }^{\left(1\right)}=\frac{Y_{1}}{\left(1+v_{1}\right)\left(1-2v_{1}\right)}\left(B_{1}+v_{1}\varepsilon_{z}^{\left(1\right)}\right)$$
(6)$$\sigma_{z}^{\left(1\right)}=\frac{Y_{1}}{\left(1+v_{1}\right)\left(1-2v_{1}\right)}\left[2v_{1}B_{1}+\left(1-v_{1}\right)\varepsilon_{z}^{\left(1\right)}\right]$$
(7)$$\sigma_{\theta }^{\left(2\right)}=\frac{Y_{2}}{1-v_{2}^{2}}\left(\varepsilon_{\theta }^{\left(2\right)}+v_{2}\varepsilon_{z}^{\left(2\right)}\right)$$
(8)$$\sigma_{z}^{\left(2\right)}=\frac{Y_{2}}{1-v_{2}^{2}}\left(v_{2}\varepsilon_{\theta }^{\left(2\right)}+\varepsilon_{z}^{\left(2\right)}\right)$$
where the superscripts 1 and 2 in parentheses indicate the corresponding values for the core and shell materials; σ is the normal stress component in the (r,θ,z) coordinate system, and ε is the corresponding strain; and *v* is the Poisson’s ratio. The uniaxial tension condition for the core/shell composite structure satisfies the following conditions: $u_{z}^{\left(1\right)}=u_{z}^{\left(2\right)}$ and $\varepsilon_{z}^{\left(1\right)}=\varepsilon_{z}^{\left(2\right)}=\varepsilon_{z}$, and the corresponding displacement of the cylindrically orthotropic material provides $u_{r}^{\left(1\right)}=\varepsilon_{r}^{\left(1\right)}r=B_{1}r$. The plane stress approximation of the shell material yields $\sigma_{r}^{\left(2\right)}=0$.

Boundary conditions that meet the continuity of displacement and traction at the interface can be presented as
(9)$$u_{r}^{\left(1\right)}\left(r_{1}\right)=u_{r}^{\left(2\right)}\left(r_{1}\right)$$
(10)$$\sigma_{\theta }^{\left(2\right)}=-\frac{r_{1}}{t}\sigma_{r}^{\left(1\right)}\left(r_{1}\right)$$
where *t* represents the shell material thickness, which is 0.34 nm for CNTs [34]. Considering the assumption of a continuous interface between the core and shell materials, the normal stress along the *z*-direction can be rewritten as
(11)$$\sigma_{z}^{\left(1\right)}=Y_{1}\left[1+\frac{2}{\left(1+v_{1}\right)\left(1-2v_{1}\right)}\frac{Y_{2}v_{1}\left(v_{1}-v_{2}\right)}{\lambda_{1}\left(1-v_{2}^{2}\right)+Y_{2}}\right]\varepsilon_{z}$$
(12)$$\sigma_{z}^{\left(2\right)}=Y_{2}\left[\frac{\lambda_{1}\left(1-v_{1}v_{2}\right)+Y_{2}}{\lambda_{1}\left(1-v_{2}^{2}\right)+Y_{2}}\right]\varepsilon_{z}$$
where
(13)$$\lambda_{1}=\frac{r_{1}}{t}\frac{Y_{1}}{\left(1+v_{1}\right)\left(1-2v_{1}\right)}$$

Accordingly, the individual normal stress components of the core and shell materials can be acquired via their material properties, that is, *Y* and *v*. Then, the total normal stress of the core/shell nanocomposite acting along the *z*-direction is obtained by
(14)$$\sigma_{z}=\frac{P_{1}+P_{2}}{A_{1}+A_{2}}$$
where *P* and *A* denote the pressure and the cross-sectional area of the material along the *z*-direction, respectively; $P_{1}=\int_{A_{1}}\sigma_{z}^{\left(1\right)}dA$ and $P_{2}=\int_{A_{2}}\sigma_{z}^{\left(2\right)}dA$.

## 4. Results and Discussion

### 4.1. Failure Mechanism of the Si/CNT Nanocomposite

Fracture behaviors of the pure [111]-oriented SiNW and Model-1 are shown in Figure 2 and Figure 3, respectively, and the diameter of the SiNW is 2.63 nm. The color indicates the atomic potential energy level. Cohesive energies of Si with a diamond cubic structure and CNTs were −4.63 and −7.40 eV, respectively (Table 1). For the [111]-oriented SiNW, nucleation of a crack along the (111) crystallographic plane was observed at *ε_z_* = 0.1825, and brittle failure occurred at *ε_z_* = 0.183. The tensile stress and strain curves (Figure 4) exhibited a failure strength of 14.3 GPa for the [111]-oriented SiNW. For Model-1, the SiNW brittle failure occurred at *ε_z_* = 0.1555. The atomic bonds of the CNT shell broke at *ε_z_* = 0.1945. SiNW failure in the case of Model-1 was noticed when the tensile stress reached 44.5 GPa.

Fracture energy (*G*) for the SiNW failure was determined from the area under the tensile stress and strain curves and was calculated using the following formula [43]:(15)$$G=\int_{0}^{\varepsilon_{z}^{c}}\sigma_{z}d\varepsilon_{z}$$
where $\varepsilon_{z}^{c}$ represents the critical strain for the SiNW failure, and the SI unit of *G* is GJ/m^3^. According to Equation (15), the *G* value for the pure [111]-oriented SiNW was 1.67 GJ/m^3^, whereas that for the SiNW failure in the case of Model-1 was 4.16 GJ/m^3^. Compared to the case of the pure SiNW, more strain energy was required for the SiNW failure to occur in the core/shell nanocomposite (Figure 5) due to the considerable load-bearing capacity of the CNTs with sp^2^-hybridized covalent bonds [20]. The fracture behavior in the case of Model-1 barely demonstrated size dependence owing to the specifically low-level crack growth resistance of the (111) crystallographic system. Intrinsic surface energy ($\gamma_{s}$) of the (111) plane has been experimentally reported to be 1.14 ± 0.15 J/m^2^ [44], which is relatively lower than those of other crystallographic systems.

Failure mechanisms of the pure [110]-oriented SiNW and Model-2 are depicted in Figure 6 and Figure 7, respectively, and the diameter of the SiNW is 2.63 nm. For the [110]-oriented SiNW, slip deformation along the (111) crystallographic plane was observed at *ε_z_* = 0.161, and the SiNW fractured when the strain exceeded 0.162. Surface energy of the (110) crystallographic plane has been experimentally reported to be 1.9 ± 0.2 J/m^2^ [44]. The image of the atomic shear invariant clearly shows slip deformation [45]. Failure strength of the [110]-oriented SiNW was measured at 14.5 GPa (Figure 8). The SiNW in Model-2 also exhibited ductile failure originating from slip deformation at *ε_z_* = 0.113. However, compared with the case of the pure SiNW, abrupt SiNW failures were prevented by the CNT shell in the case of Model-2. Strain-hardening response of the SiNW in Model-2 was observed from the stress and strain curves (Figure 8). *G* values for the [110]-oriented SiNW and Model-2 were calculated to be 1.4 and 5.3 GJ/m^3^, respectively.

Fracture behavior of the [110]-oriented SiNW demonstrated size dependence. Kang and Cai [19] showed that the [110]-oriented SiNWs with diameters in the range of 2–3 nm were fractured by slip deformation, whereas the larger SiNWs were fractured by the growth of cleavage cracks. Therefore, the diameter range of the [110]-oriented SiNWs in which a brittle-to-ductile transition (BDT) occurred was determined to be approximately 3–4 nm. The fracture behavior of the [110]-oriented SiNW with a diameter of 3.77 nm is shown in Figure 9. The longitudinal aspect ratio was fixed at approximately 3.1. According to the results reported by Kang and Cai [19], the growth of cleavage cracks along the (110) crystallographic plane leads to an abrupt SiNW failure. Interestingly, the strain hardening induced by plastic deformation was effective when the SiNWs were encapsulated by the CNTs. In the case of Model-2 (Figure 10), the LTI of the CNT shell and the diameter of the encapsulated SiNWs were (32,32) and 3.77 nm, respectively (Table 2). Contrary to the failure behavior of the pure SiNW (Figure 9), slip deformation was noticed from the core material at *ε_z_* = 0.11, and the total failure of the SiNW occurred at *ε_z_* = 0.167. The corresponding stress and strain responses are presented in Figure 8.

As is known, the fracture behavior of nanowires depends on the surface-to-volume ratio [16]. Stress distribution on the surface plays an important role in the fracture behavior of nanowires with smaller dimensions [46]. Rudd and Lee [47] have reported that the level of surface stress can influence the local structural changes of SiNWs and determine their global mechanical properties. In this study, we found that the CNT shell considerably affected the BDT of the SiNWs. Computational results were achieved for the SiNWs with diameters in the range of 1.89–6.71 nm. The failure of the [110]-oriented SiNWs encapsulated with the CNTs involved slip deformations along the (111) crystallographic plane (Figure 10). Compared to the pure [110]-oriented SiNWs, the BDT of the [110]-oriented SiNWs was not observed in the core/shell nanocomposite models. This indicates that the atomic-order interfacial interaction between the SiNWs and the CNTs significantly changes the surface elasticity of the SiNWs. The CNT shell of the core/shell nanocomposites might generate shear components that sufficiently induce plastic events in the [110]-oriented SiNWs.

### 4.2. Mechanical Properties of the Si/CNT Nanocomposite

Effective *Y* for core/shell nanocomposite can be obtained by
(16)$$Y_{\mathrm{eff}}=\frac{1}{\varepsilon_{z}}\frac{P_{1}+P_{2}}{A_{1}+A_{2}}$$

Using Equations (11), (12), and (16), $Y_{\mathrm{eff}}$ can be rewritten as follows:(17)$$Y_{\mathrm{eff}}=\frac{A_{1}}{A_{1}+A_{2}}Y_{1}\left[1+\frac{2}{\left(1+v_{1})(1-2v_{1}\right)}\cdot \frac{Y_{2}v_{1}(v_{1}-v_{2})}{\lambda_{1}(1-v_{2}^{2})+Y_{2}}\right]+\frac{A_{2}}{A_{1}+A_{2}}Y_{2}\left[\frac{\lambda_{1}(1-v_{1}v_{2})+Y_{2}}{\lambda_{1}(1-v_{2}^{2})+Y_{2}}\right].$$

Herein, *v* can be acquired by $-(r_{s}-r_{\mathrm{eq}})/r_{\mathrm{eq}}\varepsilon_{z}$, where $r_{s}$ and $r_{\mathrm{eq}}$ are the radii of the strained and unstrained materials, respectively [48]; $Y_{1}$ and $Y_{2}$ can be measured from the linear slope of the stress and strain curves for the materials 1 and 2, respectively. In this study, *Y* and *v* were obtained when a small strain was applied ($\varepsilon_{z}=0.015$).

*Y* values of the SiNWs and core/shell nanocomposites as a function of the SiNW diameter are shown in Figure 11. The aspect ratio of the SiNWs was fixed at approximately 3.1. *Y* of the CNTs was measured to be approximately 800 GPa via MD simulations based on the REBO potential (Table 1), which rarely exhibited size dependence when the CNT diameter was larger than 2 nm. The corresponding trend of CNT size dependence can be found in previously reported materials [49]. *Y* of the [111]- and [110]-oriented SiNWs slightly increased with an increase in the SiNW diameter due to the different surface-to-volume ratios, depending on the SiNW dimensions. A similar trend is reported by Rudd and Lee [47] and Kang and Cai [19]. *Y* of the core/shell nanocomposite models was substantially higher than that of the pure SiNWs owing to the high load-bearing capacity of the CNT shell [20]. Furthermore, *Y* values of the nanocomposites considerably increased with a decrease in the SiNW diameter, which was caused by the enhanced volume contribution of the CNTs.

Failure strengths of the SiNWs and the core/shell nanocomposites with different SiNW diameters are shown in Figure 12. The failure strengths of the nanocomposite models were recorded whenever the Si failure was discerned. The failure strength of the SiNWs encapsulated with CNTs was significantly higher than that of the pure SiNWs. Moreover, the failure strength of the nanocomposite models increased with a decrease in the SiNW diameter owing to the enhanced volume contribution of the CNTs. A corresponding trend was also found for the size-dependent $\gamma_{s}$ (Table 3). Theoretical fracture toughness was calculated to be $2\gamma_{s}$ based on the Griffith fracture criterion [50]. $\gamma_{s}$ for the nanocomposite models was substantially higher than that for the pure SiNWs because of the considerable CNT shell edge energy. The CNT shell edge energy has been experimentally reported to be approximately 8 J/m^2^ [51].

The values of *G*, calculated from Equation (15), corresponding to the different diameters of SiNWs are presented in Table 4. The size-dependence trend of *G* was nearly consistent with the observed surface energy trend (Table 3). The *G* values for Model-2 were slightly higher than those for Model-1. This was possibly due to the strain hardening of the encapsulated SiNWs. Note that the fracture behavior and properties of the core materials reinforced by CNTs can exhibit flaw tolerance at the nanoscale. Gao [52] suggested the concept of flaw-insensitive failure of nanocomposite materials, where a uniform rupture of nanocomposites occurs at the limiting material strength. Zhang et al. [53] also reported that the nanocrystalline graphene strip model could be fractured at the limiting strength without stress concentration at the crack tip. The effect of flaw size on the mechanical properties of the core/shell nanocomposites is not discussed in this study and will be addressed in future studies.

## 5. Conclusions

Herein, the mechanical behaviors of Si/CNT nanocomposite models under tension were analyzed via MD simulations. Interatomic forces between the Si–Si, C–C, and Si–C bonds were characterized based on the MEAM, REBO, and LJ potentials, respectively. It is concluded that the mechanical reliability of SiNWs with inherent brittleness can be significantly enhanced by encapsulating them with CNTs. The effective *Y* values and failure strengths of the Si/CNT nanocomposites were considerably higher than those of the pure SiNWs because of the high load-bearing capacity of the CNTs. Moreover, the CNT shell influenced the BDT of the [110]-oriented SiNWs. This implied that the atomic-order interfacial interaction between the SiNWs and the CNT shell possibly changed the surface elasticity of SiNWs. The findings of this study provide evidence that the core/shell nanocomposite model design can be one of the possible solutions to the problem of electrode pulverization for the development of reliable LIB electrodes.

## Figures and Tables

**Figure 1 nanomaterials-11-01989-f001:**
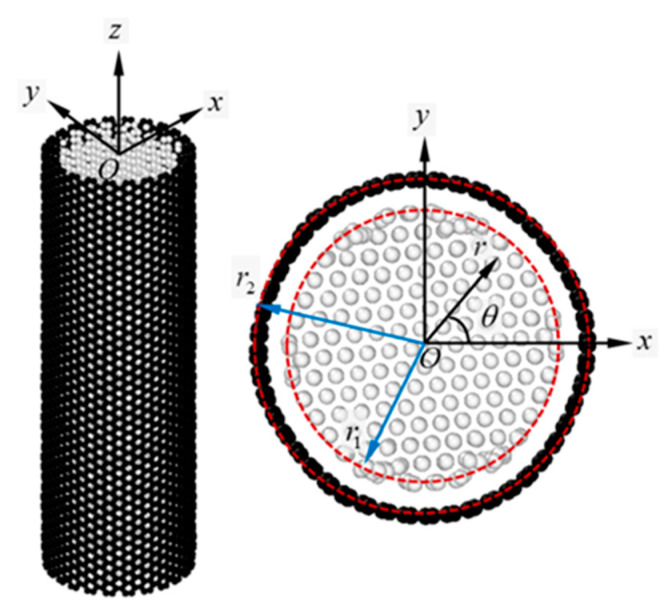
Simulation model of the Si/CNT (core/shell) nanocomposite. Axial orientation corresponds to the *z*-direction in the Cartesian coordinate system. Origin of the cylindrical coordinate system is located at *O*. $r_{1}$
and $r_{2}$, shown in the cross-sectional view, represent the radii of the core and shell materials, respectively.

**Figure 2 nanomaterials-11-01989-f002:**
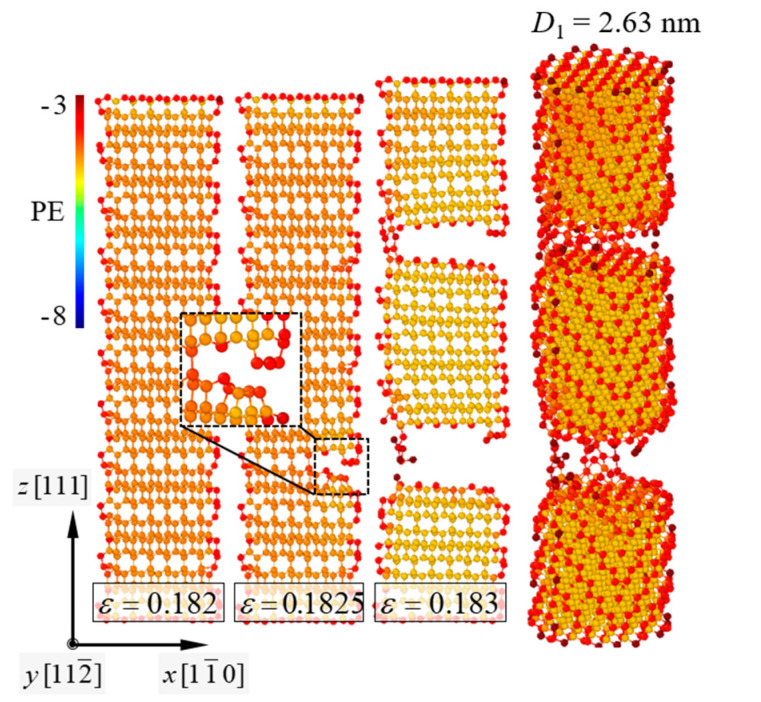
Brittle failure behavior of the [111]-oriented SiNW with *D*_1_ = 2.63 nm; *D*_1_ = $2r_{1}$. The *x*-*z* SiNW plane is depicted. PE indicates the atomic potential energy. Onset of crack propagation along the (111) crystallographic plane was observed at *ε* = 0.1825.

**Figure 3 nanomaterials-11-01989-f003:**
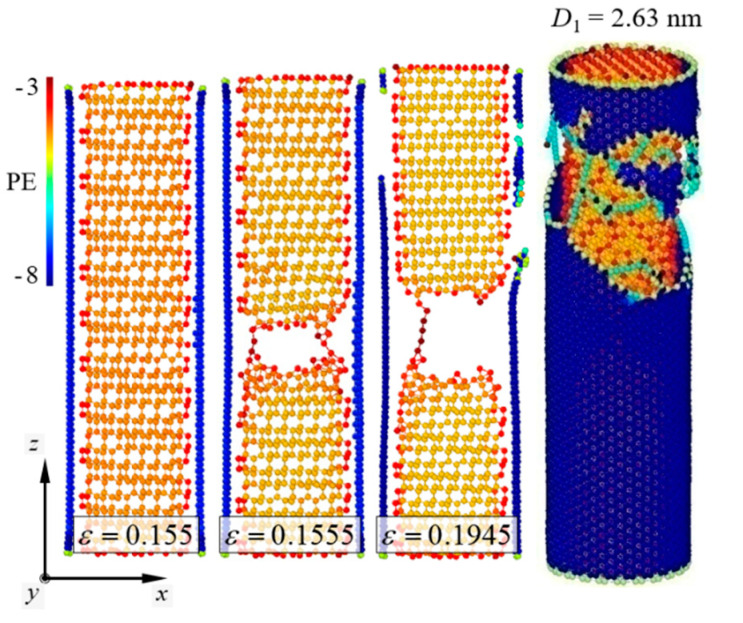
Brittle failure behavior of Model-1. *D*_1_ = 2.63 nm.

**Figure 4 nanomaterials-11-01989-f004:**
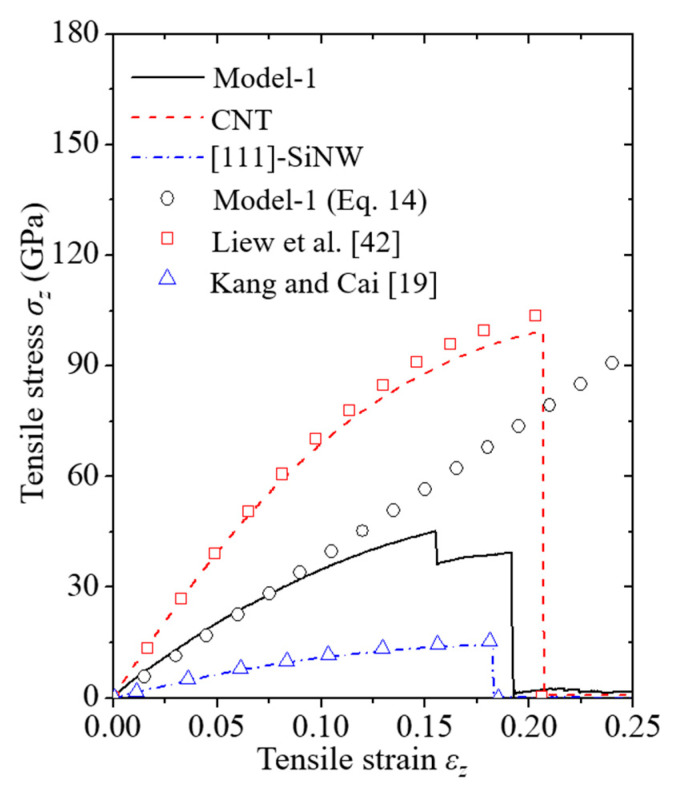
Stress and strain curves of Model-1, CNTs, and [111]-oriented SiNWs. LTI of CNTs and the diameter of SiNWs are (24,24) and 2.63 nm, respectively. Linear elastic model of the nanocomposite was obtained from the asymptotic solution of Equation (14). Curves for the CNTs and SiNWs match with the previous results reported by Liew et al. [42] and Kang and Cai [19], respectively.

**Figure 5 nanomaterials-11-01989-f005:**
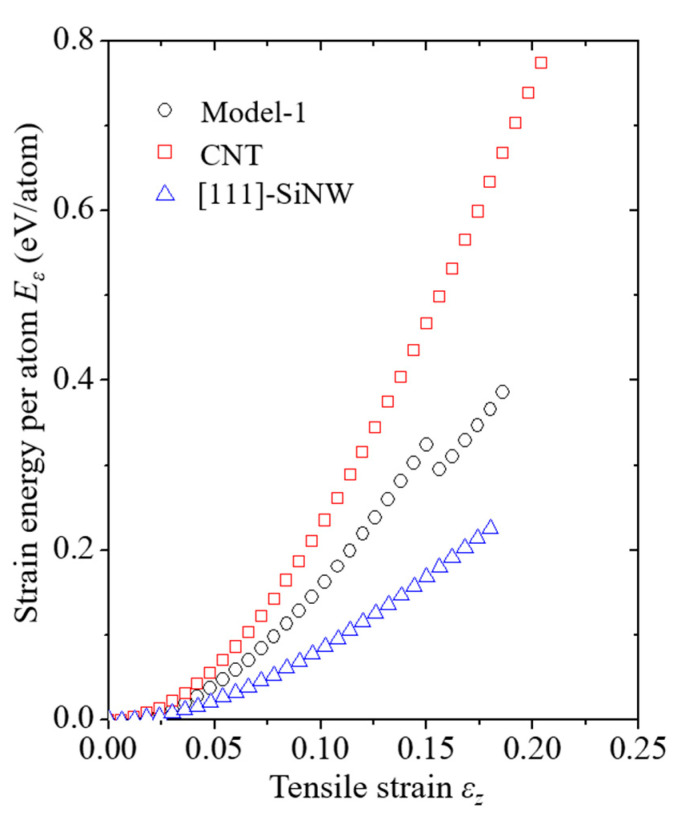
Average strain energy and strain curves of Model-1, CNTs, and [111]-oriented SiNWs. The LTI of the CNTs and the diameter of SiNWs are (24,24) and 2.63 nm, respectively. SiNW failure for Model-1 was noticed when the applied strain *ε* exceeded 0.155.

**Figure 6 nanomaterials-11-01989-f006:**
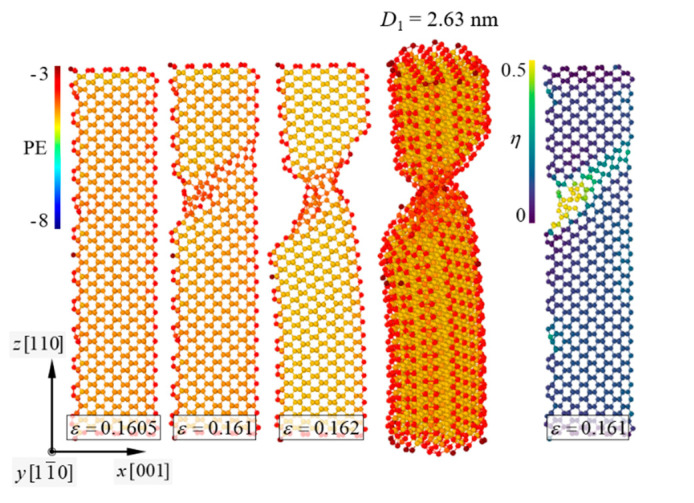
Ductile failure behavior of the [110]-oriented SiNWs with *D*_1_ = 2.63 nm. Slip deformation occurred along the (111) crystallographic plane. *η* denotes the level of the atomic shear invariant [45].

**Figure 7 nanomaterials-11-01989-f007:**
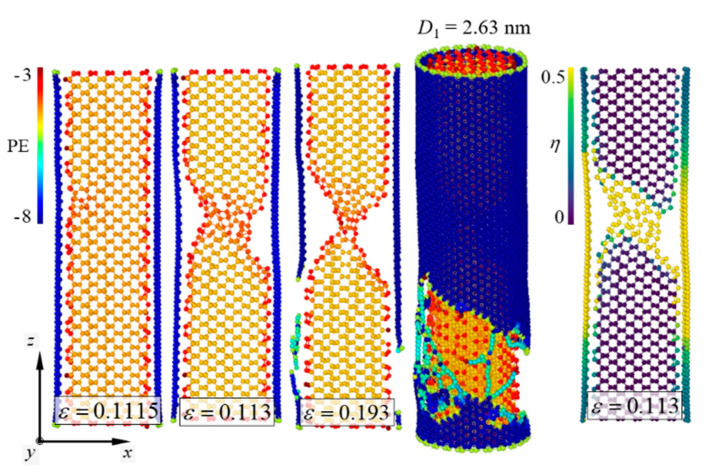
Ductile failure behavior of Model-2. *D*_1_ = 2.63 nm.

**Figure 8 nanomaterials-11-01989-f008:**
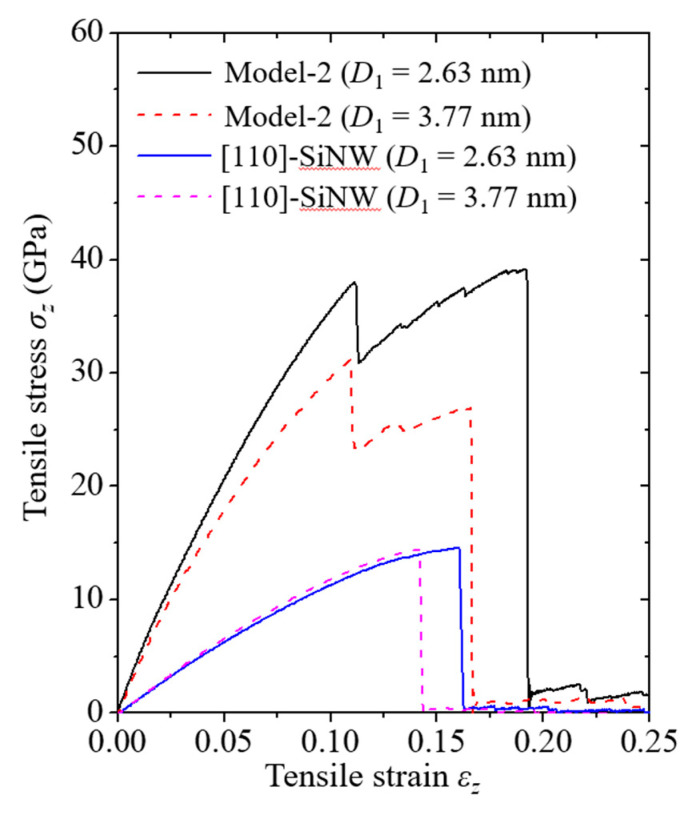
Stress and strain curves of Model-2 and the [110]-oriented SiNWs.

**Figure 9 nanomaterials-11-01989-f009:**
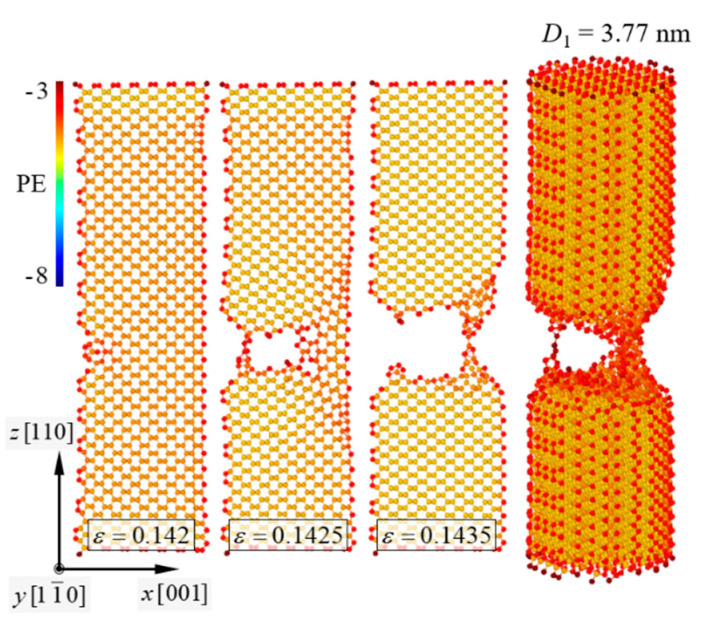
Brittle failure behavior of the [110]-oriented SiNWs with *D*_1_ = 3.77 nm. Crack propagation occurred along the (110) crystallographic plane.

**Figure 10 nanomaterials-11-01989-f010:**
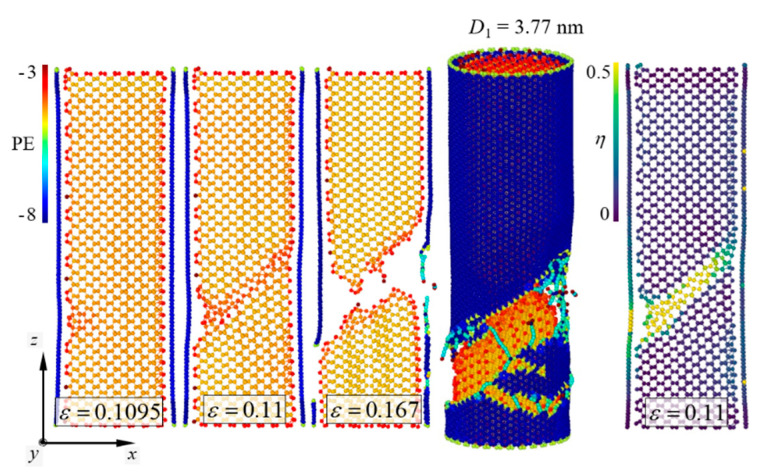
Ductile failure behavior of Model-2. *D*_1_ = 3.77 nm.

**Figure 11 nanomaterials-11-01989-f011:**
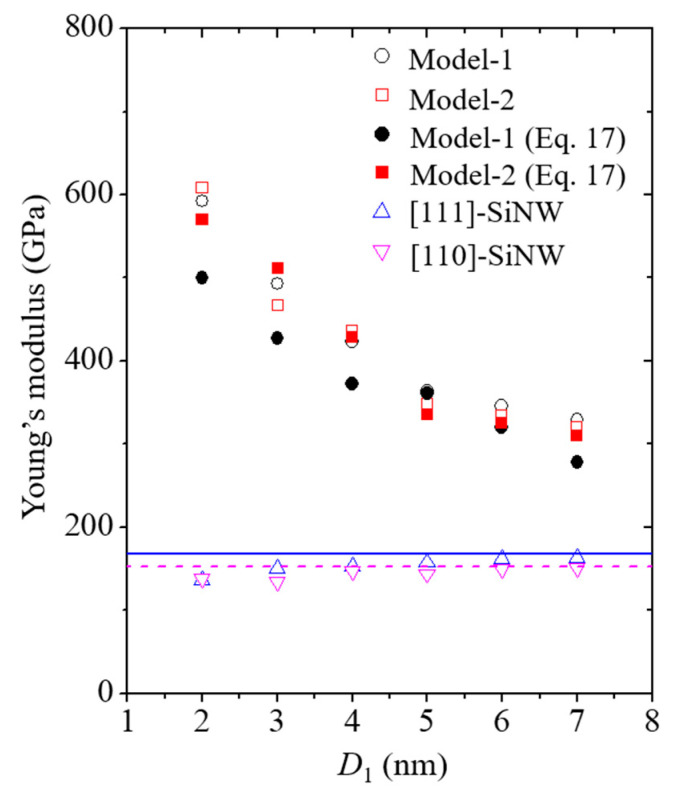
Size dependence of the Young’s moduli of the nanocomposites and pure SiNWs. Effective Young’s moduli of nanocomposites were acquired from the asymptotic solution of Equation (17). Solid blue and dashed magenta lines indicate the bulk properties of the Si single crystals with the [111] and [110] crystallographic systems, respectively, obtained from the previous study reported by Kang and Cai [19].

**Figure 12 nanomaterials-11-01989-f012:**
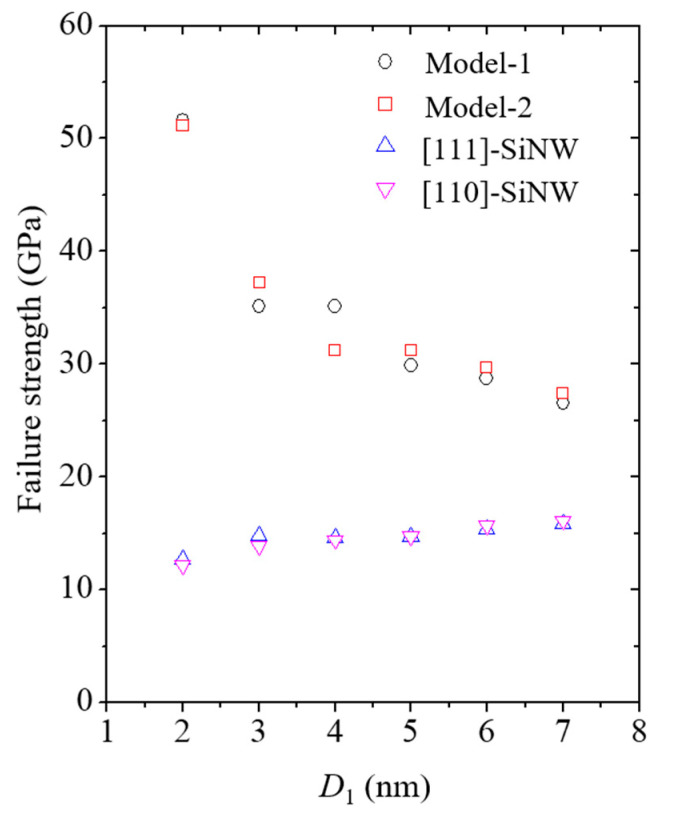
Size dependence of the failure strengths of the nanocomposites and pure SiNWs.

**Table 1 nanomaterials-11-01989-t001:** Comparison between the properties of diamond cubic Si and CNTs determined via MD calculations and experiments. *a* represents the lattice constant; $E_{\mathrm{coh}}$ indicates the cohesive energy per atom; and *Y* indicates the Young’s modulus of the [110]-oriented SiNWs with a diameter of 6.71 nm and the CNTs with an LTI of (54,54).

Property	Si–Si		C–C	
	MEAM [23]	Experiment [28,29]	REBO [26]	Experiment [28,30,31]
*a* (nm)	0.543	0.543	0.246	0.246
$E_{\mathrm{coh}}$ (eV)	−4.63	−4.71 ± 0.12	−7.40	−7.41
*Y* (GPa)	150.8	152	802.32	810 ± 410

**Table 2 nanomaterials-11-01989-t002:** Diameters of the SiNWs in the core/shell nanocomposites and the LTIs of the CNTs. *D*_1_ represents the diameter of the SiNWs. *V* is the volume fraction of the SiNWs encapsulated with CNTs. Longitudinal aspect ratio is fixed at approximately 3.1.

*D*_1_ (nm)	1.89	2.63	2.9	3.77	4.86	5.82	6.71
LTI	(18,18)	(24,24)	(25,25)	(32,32)	(40,40)	(47,47)	(54,54)
*V* (%)	60.14	66.93	69.57	74.43	77.66	80.61	81.25

**Table 3 nanomaterials-11-01989-t003:** Surface energy, $\gamma_{s}$, values of the SiNWs and core/shell nanocomposites. The SI unit of $\gamma_{s}$ is J/m^2^. *D*_1_ indicates the diameter of the SiNWs. Experimental values obtained from the study reported by Messmer and Bilello [44] represent bulk properties.

*D*_1_ (nm)	1.89	2.63	2.9	3.77	4.86	5.82	6.71	Experiment [44]
Model-1	3.84	3.13	3.15	2.70	2.38	2.21	2.07	
Model-2	3.89	3.35	3.23	2.89	2.54	2.40	2.26	
[111]-Si	0.96	1.10	1.08	1.24	1.10	1.37	1.37	1.14 ± 0.15
[110]-Si	1.18	1.31	1.16	1.32	1.29	1.34	1.35	1.9 ± 0.2

**Table 4 nanomaterials-11-01989-t004:** Fracture energy (*G*) values of the SiNWs and core/shell nanocomposites. SI unit of *G* is GJ/m^3^. *G* was calculated from Equation (15) using the stress and strain curves.

*D*_1_ (nm)	1.89	2.63	2.9	3.77	4.86	5.82	6.71
Model-1	5.86	4.16	2.11	2.91	2.28	2.31	2.00
Model-2	5.82	5.30	4.92	3.41	3.44	3.1	2.94
[111]-Si	1.33	1.67	1.67	1.71	1.42	1.51	1.62
[110]-Si	0.90	1.40	1.26	1.19	1.35	1.54	1.58

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
