# Peer review of "Mechanical Behaviors of Si/CNT Core/Shell Nanocomposites under Tension: A Molecular Dynamics Analysis"

_nanomaterials, 2021, doi:10.3390/nano11081989_

Round 1
Reviewer 1 Report
The authors present a molecular dynamics analysis of mechanical behaviors of Si/CNT core/shell nanocomposite structures. The mechanical behaviors were systematically investigated for structures with different size and also with and without CNT. However, in order to understand the results of this calculation, it is necessary to clarify the following points.
1) The state of the Si nanowire surface is not clear. What is the structure of the Si nanowire surface?
In the calculation, the surface is often terminated by hydrogen to eliminate the surface dangling bond and stabilize it. What model did you use for this calculation?
2) Si surfaces are generally covered with native oxide. I understand that adding the structure of the oxide is difficult to calculate, but what effect do you expect to have on the calculation results if the oxide is formed on the surface of SiNWs?
3) The authors calculated for SiNWs with circular shape. To minimize the surface energy of SiNWs, it is well known that some specific facets are appeared on the surfaces. The author should comment on this effect in the manuscript. It is very important to consider the effect of surface facets.
4) The author cited some references in their introduction. However the structures are not Si/CNT core/shell nanocomposite structures. Recently, research results on SiNW/CNT core/shell structures have been reported (DOI: 10.1039/d0na00098a, 10.1166/jnn.2021.19329). I recommend the authors to cite them in the Introduction.
I will recommend to publish it in Nanomaterials after you have responded to the above all points.
Reviewer 2 Report
Referee report on the manuscript entitled: „Mechanical behaviors of Si/CNT core/shell nanocomposites under tension: A molecular dynamics analysis” written by Jee S. Shim, Gi H. Lee, Cheng Y. Cui and Hyeon G. Beom
The manuscript contains theoretical results obtained on the basis of classical MD method. The main idea of the presented research is devoted to the study of mechanical behaviors of Si/CNT nanocomposite models under tension. The Authors discuss, in my opinion, an important issue related to the development of high-capacity batteries. The manuscript is well written. The idea and the presentation of the results are clear. I do not see issues, which should be corrected therefore I would like to recommend the manuscript for publication.
Author Response
We would like to thank you for your constructive comments. We glad to hear your acceptance.
Round 2
Reviewer 1 Report
As the author has responded to and revised all the items pointed out, I think it would be good to be allowed to publish on Nanomaterials.